# The Mediating Role of Gender, Age, COVID-19 Symptoms and Changing of Mansion on the Mental Health of Healthcare Workers Operating in Italy during the First Wave of the COVID-19 Pandemic

**DOI:** 10.3390/ijerph182413083

**Published:** 2021-12-11

**Authors:** Eleonora Gambaro, Carla Gramaglia, Debora Marangon, Danila Azzolina, Manuela Probo, Marco Rudoni, Patrizia Zeppegno

**Affiliations:** 1Department of Translational Medicine, Università del Piemonte Orientale, 13100 Vercelli, Italy; carla.gramaglia@gmail.com (C.G.); danila.83@live.com (D.A.); patrizia.zeppegno@med.uniupo.it (P.Z.); 2Psychiatry Unit, Maggiore della Carità Hospital, 28100 Novara, Italy; deboramarangon@libero.it; 3Department of Mental Health, ASL NOVARA, 28100 Novara, Italy; manuela.probo@asl.novara.it (M.P.); marco.rudoni@fastwebnet.it (M.R.)

**Keywords:** burnout, healthcare workers, mediation analysis, COVID-19 pandemic, mental health

## Abstract

The COVID-19 pandemic has tested the performance of hospitals and intensive care units around the world. Health care workers (HCWs) have been used to developmental symptoms, but this was especially true during the COVID-19 pandemic when HCWs have been faced with many other sources of stress and anxiety that can usually be avoided. Moreover, long-term shifts and unprecedented population restrictions have weakened people’s ability to cope with stress. The research aims to observe the dynamic interplay between burnout, depression, distress, and anxiety in HCWs working in various settings, with specific a focus on emotional exhaustion, depersonalization, and a diminished sense of personal achievement in mediating a worse mental health status during the first wave of the COVID-19 pandemic in Italy. We performed a mediation analysis, which resulted in a strong correlation among depression, psychological distress, health perception and anxiety, and the impact of job burnout on anxiety, depression, and distress. Gender seemed to have a strong correlation with burnout, anxiety, and distress; the impact of the COVID-19 pandemic on Quality of Life seemed to affect anxiety and depression; the possible changes in job tasks and duties (intended as a change in work area or location and role change)influenced depression and job burnout. Encouraging supportive and educational strategies would be recommended to policymakers.

## 1. Introduction

The WHO Emergency Committee declared a global health emergency on 30 January 2020, due to COVID-19 (coronavirus disease 2019; previously 2019nCoV) outbreak disease [1,2]. The cumulative number of global confirmed cases reported is now 254,847,065, including 5,120,712 deaths, reported to WHO and the cumulative number of deaths is 5,120,712 million. A total of 7,370,902,499 vaccine doses have been administered (data reported at 18 November 2021 Figure 1) [2]. COVID-19, as an unknown disease, requires in-depth studies and observations on the existence of the virus, thus posing itself as a new challenge for the scientific community. To contain the disease, develop prevention and treatment strategies, active loco-regional to international cooperation is necessary [3].

The pandemic of COVID-19 would force a re-definition of vital support personnel, with acknowledgement of all healthcare workers (HCWs) contributions and adequate education, defense, and compensation [4].

According to the COVID-19 Task Force of the Department of Infectious Diseases and the IT Service of the Italian Higher Institute of Health, the cases in the general population were 4,893,529 (49% of whom males, 51% of whom females, with an average age equal to 45 years), the cases among healthcare workers (HCWs) were 148,045, deaths were 132,413 and cases cured were 4,487,286 (data updated as of 18 November 2021) [5].

The COVID-19 pandemic has tested and, in many cases, surpassed hospital and intensive care unit (ICU) capabilities around the world [5]. Despite fatigue, personal risk of infection, fear of transmission to family members, sickness or death of friends and colleagues, and the loss of many patients, HCWs is a population of workers already used to developing anxiety, depression, burnout, insomnia, moral distress, and post-traumatic stress disorder [6], and have continued to provide care for patients. Furthermore, they have also had to cope with a slew of other issues as well as long shifts coupled with unprecedented population limits, such as personal isolation, have harmed people’s ability to cope [7]. Loneliness has been exacerbated by working remotely and being shunned by community members [8].

Many HCWs travelled to new places of work as the pandemic progressed, often thrust into the pandemic ICU environment, with inadequate skills and training [9]. Hospital-based HCWs have had to work long hours wearing bulky and uncomfortable personal protective equipment (PPE) [7]. 

The treatment of COVID-19 patients with chronic comorbidities has been particularly complex due to both the lack of funding and specific COVID-19 therapies [10]. Some HCWs have been faced with difficult decisions regarding resource rationing and withholding resuscitation or ICU admission causing emotional and ethically fraught dilemmas [11]. Moreover, because of the COVID-19 pandemic, surgeries or other life-saving treatments were cancelled or postponed, leading to anguish among recovered COVID-19 patients, an emotional experience shared by the attending physicians [12,13].

HCWs’ preparation (e.g., medical students, residents, and allied health learners) was also disrupted, resulting in tuition payments being lost, missed learning opportunities, missed tests, and possibly delayed certification [7,14]. COVID-19 was particularly relevant for female HCWs, where COVID-19 has had a disproportionately negative effect. Women make up 70% of the global health and social care workers, placing them at risk of illness and the variety of physical and mental health issues that come with their roles as health practitioners and caregivers [7]. Women have had to balance their professional obligations with their family’s needs, including childcare, homeschooling, elder care, and home care [15,16]. Due to these commitments, women’s academic productivity was lower than men’s, as shown by the fact that fewer women were part of the cohort that generated new information about the pandemic [17]. 

Although it is known that extreme burnout syndrome affects up to 33% of critical care nurses and up to 45% of critical care physicians under normal working circumstances [18], nevertheless, data on impact of COVID-19 pandemic is not available. Surely, increased job pressures and little influence over the work environment, as well as the trauma of caring for critically ill patients, are significantly exacerbating factors for poor mental health among HCWs [19]. 

Burnout is a multi-faceted reaction to physiological, mental, or interpersonal work stressors that can lead to psychological issues, increased suicide, and drug abuse among HCWs [20]. Emotional exhaustion (EE), depersonalization (D), and a diminished sense of personal achievement (PA) are all symptoms of this condition [21]. A variety of factors can affect burnout risk, but encouraging mental health in policies, reducing HCW workload, mitigating job-related stressors, and favoring a healthy work environment can all help to prevent or reduce burnout [22].

In a previous study using an online survey [23], we evaluated the mental health effects of the first wave of the COVID-19 pandemic on HCWs in North-Eastern Piedmont, a high-risk region in Italy. Our study analyzed HCWs from various settings (hospital and community healthcare facilities, emergency, and non-emergency services), including also HCWs not directly involved in the treatment of patients affected by the COVID-19 disease, to observe the dynamic interplay between burnout, depression, distress, and anxiety in HCWs working in various settings, with a specific focus on EE, D, and a diminished PA in mediating a worst mental health status. 

Based on the research findings reported above, the objective of the study was to explore: (1) whether a relationship between job burnout and perceived stress, anxiety, and depression exists among HCWs; (2) whether socio-demographic and anamnestic characteristics could act as a mediator in the relationship between job burnout and perceived stress, anxiety, and depression.

## 2. Materials and Methods

The study protocol was approved by the local Ethical Board (Comitato Etico Interaziendale di Novara, Protocollo 534/CE, Studio n. CE 82/20, approved on 11 May 2020). The survey was implemented with the REDCap platform and e-mailed at the end of the first wave of the COVID pandemic emergency crisis period (in June 2020) on behalf of the human resources offices in charge of the healthcare institutions detailed below, who have access to the mailing lists including the institutional e-mail contacts of all HCWs employees. The procedure for the implementation and diffusion of the survey have been already described in detail elsewhere [23].

The online survey presented the objectives of the research; HCWs were required to give their informed consent to participate. 

The first part of the online survey included general information, questions about the professional role and possible changes in job tasks and duties (intended as a change in work area or location and role change) during the peak of the pandemic. Regarding the reality of Novara, Italy, all ordinary leave was suspended; resources were redistributed by reducing the number of beds and therefore the number of personnel assigned to ordinary departments to allocate them to crisis departments. The territorial network was also strengthened to monitor cases in isolation/quarantine and to identify possible contacts of ascertained cases. Some private health facilities have also been used to treat patients with COVID-19. The medical staff was reorganized in the various departments so that structured doctors with expertise in the management of respiratory patients were placed at the forefront, with a coordinating role: internists, pulmonologists, infectious diseases, anesthetists. These were supported 24 h a day by colleagues from the remaining support departments. Doctors in specialist training were also located to support wards with COVID-19 patients, carrying out back-office activities. The reorganization also provided for the redistribution of nursing staff and social and health workers.

Standardized and validated self-administered measures were used for the assessment of burnout (Maslach Burnout Inventory—Human Services Survey for Medical Personnel-MBI-HSS MP) [24], overall health perception (General Health Questionnaire-12 Items-GHQ) [25], distress perceived because of stressing life events (Impact of Event Scale-IES) [26], depression (Beck Depression Inventory-BDI) [27] and anxiety (Beck Anxiety Inventory-BAI) [28].

MBI-HSS MP is a 22-items scale, with each item scored on a 7-point (ranging from 0, “never,” to 6 “every day”). It evaluates individuals’ experience of occupational burnout in individuals who work with people (human services and medical professionals), with three components: EE, D, and reduced PA. Each subscale score can then be coded as “low,” “average” or “high” according to the scoring key, and is considered separately from the other, without combining into a single, total score. GHQ was developed for non-clinical populations to detect a wide range of disorders, and specifically the anxiety/depression spectrum; it is a valid and reliable instrument across cultures. The items are rated on a 4-point scale (“less than usual,” “no more than usual,” “rather more than usual,” “much more than usual”) offering a total score ranging from 0 to 36 points, with higher scores indicating worse mental well-being. IES consists of 15-items, rated on a 4-point scale according to how often each has occurred in the past 7 days (0 = not at all; 1 = rarely; 3 = sometimes; 5 = often). Besides the IES total subjective stress score, two subscales are identified: one for intrusive symptoms (intrusive thoughts, nightmares, intrusive feelings, and imagery; seven items, scores ranging from 0 to 35), and one for avoidance symptoms (numbing of responsiveness, avoidance of feelings, situations, ideas; eight items, scores ranging from 0 to 40). The IES has also displayed the ability to discriminate a variety of traumatized groups from non-traumatized groups. Even if, the questionnaire evaluating trauma used in this study was not the Impact of Event Scale with modifications for 2019-nCOVID (IES-COVID19), the protocol was specified to answer the IES questions considering “event”, everything that was related to the 2019-nCOVID pandemic. BAI is a 21-item scale for the assessment of anxiety symptoms severity. Each item is rated on a 4-point scale (from 0 = not at all to 3 = severely, can barely stand it), focusing on the past week experience. The final score, obtained from the sum of the individual items, is between 0 and 63 (≤21 = minimum level of anxiety; 22–35 = medium level of anxiety; ≥36 = high level of anxiety). BDI is a 21-item self-report inventory measuring the severity of depression. Each item is scored on a 4-point scale. In this inventory, the higher is the total score, the more severe is depression; standardized cutoff values are the following: 0–13 = minimal depression; 14–19 = mild depression; 20–28 = moderate depression; 29–63 = severe depression.

In our sample of HCWs, four main subgroups could be identified: medical doctors/physicians, residents in training (meaning graduated medical doctors attending specialization schools), nurses and “others” (this group included participants who did not fit any of the previous categories, such as psychologists, social workers, radiology and laboratory technicians, educators).

### Statistical Analysis

Data have been synthesized in terms of absolute and relative frequencies for the categorical variables and as median and interquartile range (Q1–Q3) for the continuous variables.

Mediation analysis has been carried out via Structural Equation Model (SEM) computation. 

All endogenous dependent variables are quantitative; the Pearson correlation matrix with associated densities and histograms have been reported for these variables.

The biserial (polyserial) correlations have been reported for the continuous and binary (ordinal) variables and Tetrachoric (or Polycoric) correlations for binary (ordinal) variables. The intercepts in the SEM were set to zero. The dichotomous variables used in the SEM model (gender, change in habits during the pandemic, etc.) are exogenous (independent). Those variables have been recorded as a dummy (0/1) variable like in a classic regression model as suggested in the literature [29]. The age in classes variable is ordinal and exogenous, the encoding scheme reflects the order (say, 1,2,3,…) and has been treated like any other (numeric) covariate [29].

The computations have been conducted with R 3.4.2 [30] with the lavaan [29] packages (R Foundation for Statistical Computing, Vienna, Austria).

## 3. Results

### Description of the Sample

The online survey was e-mailed to 2422 HCWs and completed by 897 (37%) respondents. 

In total, only 653 out of these 897 (73%) completed the questionnaires in full. The 244 incomplete records were thus excluded from the statistical analyses. 

210 HCWs were male (32.2 %), 443 HCWs were female (67.8 %). 92 HCWs were aged 18–29 years (14.1%), 189 HCWs were aged 30–39 years (51%), 145 HCWs were aged 40–49 years female (22.2%), 227 HCWs were aged ≥50 years (34.8%). 159 HCWs were single/divorced/widow (24.3%), 413 HCWs were married/cohabitant (63.2%), 81 HCWs in a stable relationship (12.4%), 358 HCWs had children (54.8%), 295 HCWs did not have children (45.2%). 

Regarding the MBI-HSS MP scale, the median EE is 18 (moderate), D is 18 (high) and PA is 32 (high), indicative of moderate-high levels of burnout. The median IES scale is 19 (mild subjective stress), the median GHQ scale is 18 (perception to have some health problems). The median BAI and BDI scale are 8 (mild anxiety, low depression).

Doctors/physicians comprised 286 HCWs (43.8 %), 99 HCWs were residents in training (15.2 %), 137 HCWs were nurses (21.0 %), 131 HCWs were other professionals (20.1 %). Positive test results for a COVID-19 swab were 89 HCWs (13.6%),125 had COVID-19 related symptoms (19.1%). 556 HCWs did not have COVID-19 related health problems (85.1%). A total of 322 HCWs modified their job due to the COVID-19 pandemic (49.3%), 331 HCWs did not modify their job due to the COVID-19 pandemic (50.7%). HCWs who had someone close to them test positive to a COVID-19 swab were 454 (69.5%), 199 HCWs did not have someone close test positive to a COVID-19 swab (30.5%). A total of 43 HCWs modified family habits for fear of infecting a loved one (6.6%), 525 HCWs did not modify family habits for fear of infecting loved one dear (80.45), 85 HCWs did not answer the question related to modification of family habits for fear of infecting loved one dear (13.0%). All results are shown in Table 1. 

The statistical significance of the mediating effect was confirmed by the Sobel test. The SEM yielded a good fit to the observed data indicating the direct pathway from job burnout and perceived stress, anxiety or depression and the indirect pathway, which was mediated by other characteristics. As shown in Figure 2 and Table 2, the direct effect of perceived stress on job burnout was estimated in the model (the model fit of the data χ 2/df < 5, *p* < 0.05), which was found to be not statistically significant and positive (β = 0.28); there were statistically significant effects of perceived stress (IES) on both anxiety (β = 0.61) and depression (β = 0.58). Moreover, there seems to exist statistically significant effects of depression on anxiety (β = 0.78) and health perception on depression (β = 0.68) and anxiety (β = 0.56). The coefficients of perceived stress on job burnout were significantly reduced (β = 0.28) as also for health perception on perceived stress (β = 0.44) and job burnout (β = 0.36).

In the second phase of mediation analysis (Figure 3 and Table 3), the three variable scales of the MBI-HSS (MP) questionnaire were considered individually, i.e., EE, which measures feelings of being emotionally overextended and exhausted by one’s work, D that measures an unfeeling and impersonal response toward patients, and PA those measures feelings of competence and achievement in one’s work. From this analysis it emerged that there were statistically significant effects of EE on D (β = 0.58), PA (β = 0.14) anxiety (β = 0.53) and depression, and less significant effects on psychological distress (β = 0.32) and health perception (β = 0.46). Regarding D, it showed lower statistically significant effects on anxiety (β = 0.37), depression (β = 0.40), psychological distress (β = 0.19), and health perception (β = 0.27), but greater statistically significant effects on the reduction of PA, indicative of higher burnout (β = −0.01).

Finally, effects of a low PA were particularly significant on perceived stress (β = 0.02), but also on anxiety (β = 0.14), health perception (β = 0.15), and depression (β = 0.18).

Four covariates were included in the mediation analysis: age, gender, COVID-19 related symptoms, and changing of mansion due to the COVID-19 pandemic. As shown in Figure 4 and Figure 5, Table 4, Table 5 and Table 6 emerged as showing that the covariates were not very correlated with each other. Moreover, a statistically significant effect emerged of gender on anxiety (β = 0.656) and psychological distress (β = 0.855), and of changing of mansion due to the COVID-19 pandemic on Maslach total (β = 0.716), health perception (β = 0.538), depression (β = 0.582), and perceived stress (β = 0.502). Finally, it emerged that COVID-19 related symptoms had statistically significant effects on anxiety (β = 0.520), but weaker ones on depression (β = 0.495), perceived stress (β = 0.319), and health perception (β = 0.397).

Considering EE, D, and PA singularly, statistically significant effects were found as follows: age categories on low PA (β = −0.217); gender on EE (β = 0.698) and on low PA (β = 0.164); changing of mansion due to the COVID-19 on EE (β = 0.556) and on low PA (0.057), health perception (β = 0.538), depression (β = 0.582), perceived stress (β = 0.502).

The SEM estimated RMSEA fit is equal to 0.075 indicating a suitable model fit.

## 4. Discussion

The 2019-nCOVID pandemic as a public health emergency has presented healthcare systems with remarkable challenges. The current research expands the findings of a previously published study, to deepen the understanding of the mental health effects of the 2019-nCOVID pandemic on HCWs from North-eastern Piedmont, Italy. In the first work [23], we evaluated singularly burnout, anxiety, depression, distress, observing higher degrees of burnout (in particular D and PA) in females, in HCWs aged <30 years, in those exposed to changes in their working habits and their families’ behavior, and in trainees. Moreover, lower ranges of anxiety and depression than those reported in the literature were found.

To our knowledge, this study was the first to explore the relationship among job burnout, depression, anxiety, perceived stress, health perception in Italian HCWs, examining the possible role of the following factors as mediators of the aforesaid relationships: gender, age categories, COVID-19 related symptoms, changing of mansion due to the COVID-19 pandemic.

We wanted to investigate, by utilizing mediation analysis, whether a variable (i.e., mediator) adjustment regarding an impartial variable, in turn, affects a structured variable. Moderation evaluation, however, investigates whether the statistical interplay between impartial variables expects an established variable, with a specific interest in the role of the three scales of job burnout.

Only 37% of HCWs responded to the emailed survey. The low response rates highlighted a possible lack of interest in participating in the study in a tragic and unexpected historical period such as the pandemic, which has led to an upheaval in family and work habits; the length of the survey; the absence of certainty of protection of privacy and confidentiality are all factors. As described in a previous study [31], the COVID-19 pandemic has led to survey fatigue characterized by non-response, with a consequent decrease of response rate during the pandemic. This could be explained by the fact that during the COVID-19 pandemic, the number of surveys created and disseminated has increased significantly with the consequence that HCWs may feel overwhelmed with the number of survey requests, also due to the great increase in social media dissemination during the COVID-19 pandemic that can contribute to the illusion that survey requests are omnipresent.

The results of this study highlighted statistically significant effects of perceived stress and health perception on both anxiety and depression, and effects of depression on anxiety. As for job burnout, we found statistically significant effects of EE on D, PA, anxiety, depression, and health perception. The statistical correlation between D and a low PA was highlighted, indicative of higher burnout. Finally, effects of a low PA were significantly correlated with perceived stress, anxiety, health perception, and depression.

Four covariates were included in the mediation analysis: age, gender, COVID-19 related symptoms, changing of mansion due to the COVID-19 pandemic.

Findings suggested the following: age categories impact on low PA; gender impacts on anxiety and psychological distress; changing of mansion due to the COVID-19 pandemic impacts on health perception, depression, perceived stress, EE, and low PA; COVID-19 related symptoms have statistically significant effects on anxiety.

Using the structural equation model (SEM), Song et al. [32] described that both tension and poor rest showed associations with job burnout among Chinese nurses. The SEM analysis confirmed the direct pathway from perceived stress to burnout (β = 0.69, *p* < 0.05) and the indirect pathway mediated by sleep quality (β = 0.56). There existed statistically significant effects of sleep quality on both perceived stress (β = 0.48) and job burnout (β = 0.29). Nonetheless, in our study, we did not analyze the quality of the sleep–wake rhythm and our sample included different groups of HCWs; therefore, the possibility to compare our results to those by Song and coworkers is limited. Notwithstanding these limitations, it is true that a correlation between perceived stress and job burnout has been found in both works, even though the effect of low PA (high burnout) on perceived stress was more evident in this study.

A Turkish study [33] aimed to examine the mediating role of optimism and social relationships on the development of burnout among HCWs during the COVID-19 pandemic. Women reported greater strain from the COVID-19, greater emotional exhaustion, and fewer social relationships. HCWs with COVID-19 disease reported less optimism. The findings suggested that stress and anxiety not only had a direct effect on increasing COVID-19 burnout but also had an indirect effect on it through a decrease in positive outlook and social connections. Even if our work did not specifically investigate social relationships, it was observed that changing of mansion due to the COVID-19 pandemic had an impact on health perception, depression, perceived stress, and burnout (high EE and low PA); moreover, in our sample, HCWs with COVID-19 related symptoms reported higher levels of anxiety symptoms.

A national cross-sectional survey conducted in the U.S. analyzed the prevalence and correlates of stress and burnout among HCWs during the COVID-19 pandemic [34]. Higher Summary Stress Score (SSS) which included stress, fear of exposure, anxiety/depression, and workload were highlighted among nursing assistants, medical assistants, social workers, inpatients, women, and black individuals; moreover, the results appeared to be related to workload and mental health, and the SSS score was lower when health professionals felt valued. The workload in our study was objectified through the change of mansion due to the COVID-19 pandemic, which led to a worsening of perceived health, an increase in distress, depression, and job burnout (high EE, low PA). Gender also appears to impact anxiety symptoms and psychological distress as found in the previously cited study.

In a Portuguese study analyzing the mediating role of psychological resilience of HCWs during the COVID-19 pandemic on burnout and depression [35], the outcomes revealed that clinical depression had a direct guided effect on the individual, job- and also patient-related burnout, as in our study, where it found the correlation between depression and EE, in addition to the strong correlation with anxiety. Moreover, Serrão et al. [35] also observed a small indirect impact of depression on burnout, mediated by resilience; resilience played a partial mediating role between anxiety as well as all job burnout measurements.

One study conducted during the first COVID-19 pandemic peak period to analyze the burnout status of Italian HCWs [36] showed that a substantial part of the sample scored over the clinical levels of depression (57.9%), anxiety (65.2%), post-traumatic symptoms (55%), and also burnout (25.61%). The burnout variation highlighted in the study by Conti et al. seemed to be independently affected by working on the front line, being doctors, experiencing reductions in mental health, as well as higher levels of post-traumatic stress disorder symptoms, in line with the results of our work. We found that the EE and PA scales of MBI-HSS MP, correlated with anxiety and depression, while D showed a lower impact on them. Moreover, the EE and PA scales seemed to have an impact on HCWs’ health perception.

### Strengths and Limitations

Among the strengths of our study was the sample size and the use of validated questionnaires to investigate burnout with anxiety, depression, distress symptoms, and overall mental health. Moreover, our survey was sent to frontline and non-frontline HCWs, recruited both from the health facility as well as from extra-hospital settings, allowing for an in-depth understanding of the pandemic that has had a unique effect on HCWs. Furthermore, data were gathered about socio-demographic, working habits-related, and pandemic-related variables.

Nonetheless, our lookup has some boundaries which need to be underscored. We gathered data solely from a single center in Piedmont, a high-risk though restrained area, in Italy. This is a cross-sectional study, and in accordance with the design, it is challenging to derive causal relationships. As all comparable research in this field, regrettably, goal records about preceding psychiatric issues had been no longer reachable and we did not ask for information about preceding psychiatric history, which may have biased the results we found. We had no availability of preceding measures on the psychological variables investigated, nonetheless, it is probable that, for burnout and perceived time-honored health, ratings worsened throughout the present-day pandemic. It should also be underlined that our study is the first that analyzes the correlation among the scales of MBI-HSS MP with others. More specifically, it was highlighted a high correlation between EE and D was highlighted as well as D having a statistically significant effect on the reduction of PA.

Moreover, as in different comparable studies, we used online self-report instruments that are much less inclusive and less precise than an assessment interview performed by a skilled clinician. Finally, the validated scale for the contrast of stress signs and symptoms (COVID-19 IES) [37] had not been used because it was not yet available. Nonetheless, the follow-up of the sample will include COVID-specific measures.

## 5. Conclusions

The COVID-19 pandemic has exposed the general population to challenges never seen before, including restriction of social relationships, and changes in individual and family habits. While supported by institutional and government leadership, the spirit of collaboration, the celebration of saved lives and the public recognition of their relevance, the HCWs have displayed high levels of distress, anxiety, emotional exhaustion, which contributed to increasing feelings of loneliness and the deterioration in their mental health [7].

Our study showed a particularly strong correlation among depression, psychological distress, health perception and anxiety, and the impact of job burnout (high EE) on anxiety, depression, and distress. Gender seemed to have a strong correlation with burnout (High EE), anxiety and distress, while the impact of COVID-19 pandemic on Quality of Life (QoL) seemed to affect anxiety and depression while changing of mansion due to the COVID-19 pandemic influenced depression and job burnout (high EE).

The long-term influence on the well-being of health care workers has yet to be established. During the COVID-19 pandemic, HCWs experience increased emotional stress and anxiety and, in many cases, depression and mental illness.

Encouraging supportive, motivational, protective, and educational strategies would be recommended to policymakers and managers [16].

Identifying the common mental distress related to the care of people with COVID-19, through the analysis of mediating factors that contribute to increased psychological distress and job burnout, would allow destigmatizing mental illness among HCWs, finalizing prevention, and treatment strategies for this population.

## Figures and Tables

**Figure 1 ijerph-18-13083-f001:**
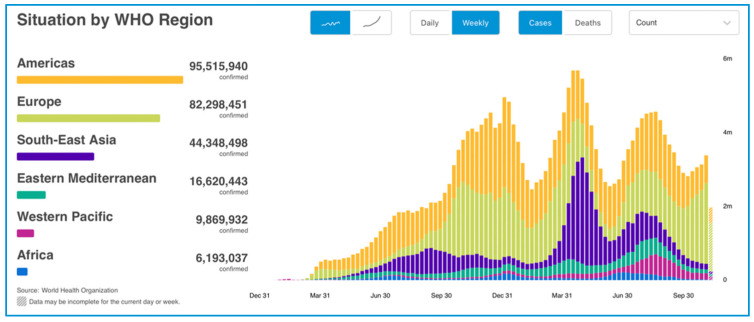
COVID-19 cases reported weekly by WHO Region, and global deaths, as of 18 November 2021 tried by World Health Organization 2021.

**Figure 2 ijerph-18-13083-f002:**
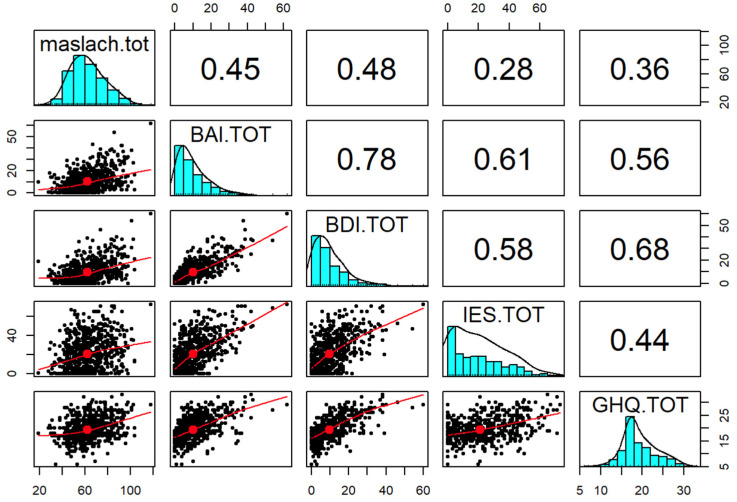
Observed Pearson correlations among Medical Personnel (MBI-HSS MP), Beck Depression Inventory (BDI-II), Beck Anxiety Inventory (BAI), Impact of Event Scale (IES) and General Health Questionnaire 12 Items (GHQ-12).

**Figure 3 ijerph-18-13083-f003:**
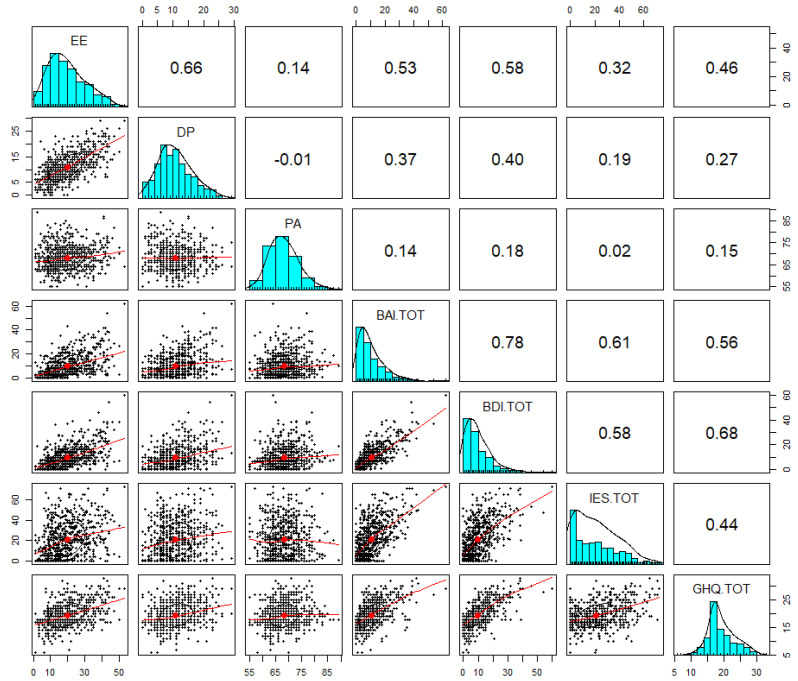
Observed Pearson correlation among Emotional Exhaustion (EE), Depersonalization (DP), Personal Accomplishment (PA), Beck Depression Inventory (BDI-II), Beck Anxiety Inventory (BAI), Impact of Event Scale (IES) and General Health Questionnaire 12 Items (GHQ-12).

**Figure 4 ijerph-18-13083-f004:**
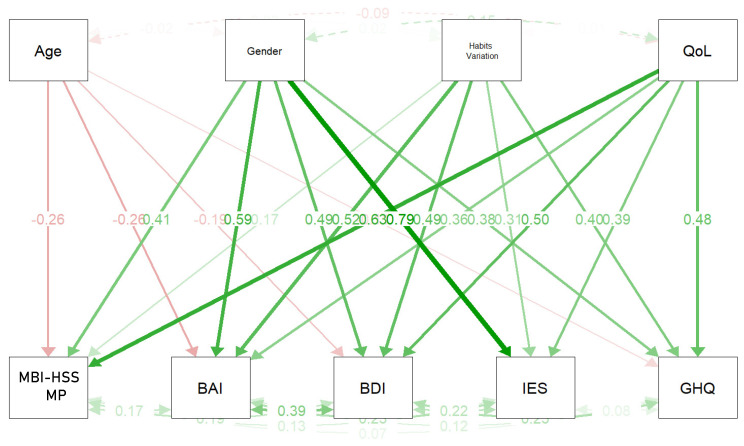
Mediation analysis with covariances (Age categories, Gender, Age categories, Gender, COVID-19 related symptoms, Changing of mansion due to the COVID-19 pandemic). Correlations among Medical Personnel (MBI-HSS MP), Beck Depression Inventory (BDI-II), Beck Anxiety Inventory (BAI), Impact of Event Scale (IES) and General Health Questionnaire 12 Items (GHQ-12).

**Figure 5 ijerph-18-13083-f005:**
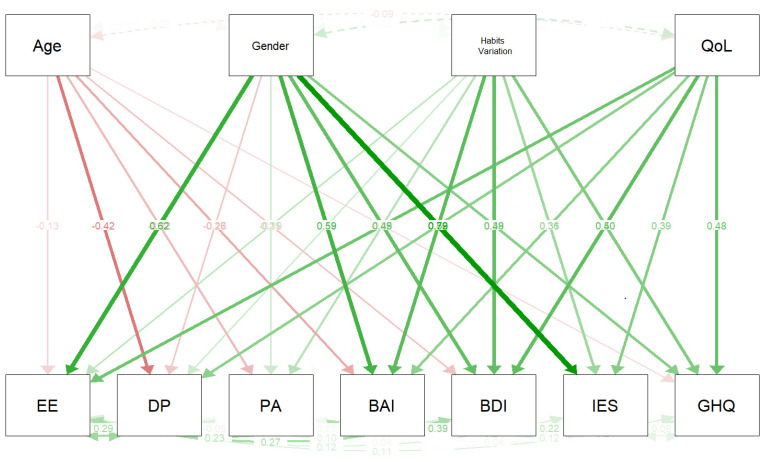
Mediation analysis with covariances (Age categories, Gender, Age categories, Gender, COVID-19 related symptoms, Changing of mansion due to the COVID-19 pandemic). Legend: Correlations among Emotional Exhaustion (EE), Depersonalization (DP), Personal Accomplishment (PA), Beck Depression Inventory (BDI-II), Beck Anxiety Inventory (BAI), Impact of Event Scale (IES) and General Health Questionnaire 12 Items (GHQ-12). QoL = Impact of COVID-19 pandemic on Quality of Life (QoL).

**Table 1 ijerph-18-13083-t001:** Descriptive data of the sample, including socio-demographic and work-related variables. The median and quartiles (Q1–Q3) have been reported for the continuous variables and the absolute with relative % frequencies for the categorical ones.

Variables				*n* Median	% Q1–Q3
**MBI-HSS MP**		**EE**	Emotional Exhaustion	18	11–26
		**D**	Depersonalization	10	7–14
		**PA**	Personal Accomplishment	32	28–36
**IES**			Impact of Event Scale	19	6–33
**GHQ-12**			General Health Questionnaire	18	17–22
**BAI**			Beck Anxiety Inventory	8	4–14
**BDI-II**			Beck’s Depression Inventory	8	3–14
**Gender**			Male	210	32.2%
			Female	443	67.8%
**Age categories**			18–29 years	92	14.1%
			30–39 years	189	28.9%
			40–49 years	145	22.2%
			≥50 years	227	34.8%
**Marital status**			Single/divorced/widow	159	24.3%
			Married/cohabitant	413	63.2%
			In a stable relationship	81	12.4%
**Children**			Yes	358	54.8%
			No	295	45.2%
**Working categories**			Doctors/Physicians	286	43.8%
			Residents in training	99	15.2%
			Nurses	137	21.0%
			Others *	131	20.1%
**Positivity to COVID-19 swab**			No	564	86.4%
			Yes	89	13.6%
**COVID-19 related symptoms**			No	528	80.9%
			Yes	125	19.1%
**Health problems not related to COVID-19**			No	556	85.1%
			Yes	97	14.9%
**Changing of mansion due to the COVID-19 pandemic**			No	89	13.6%
			Yes	564	86.4%
**Job modification due to the COVID-19 pandemic**			No	322	49.3%
			Yes	331	50.7%
**Someone of dear positive to COVID-19 swab**			No	454	69.5%
			Yes	199	30.5%
**Modification of family habits for fear of infecting loved one dear**			No	43	6.6%
			Yes	525	80.4%
			No answer	85	13.0%
**Gender**			Male	210	32.2%
			Female	443	67.8%
**Age categories**			18–29 years	92	14.1%
			30–39 years	189	28.9%
			40–49 years	14	22.2%
			≥50 years	227	34.8%
**Marital status**			Single/divorced/widow	159	24.3%
			Married/cohabitant	413	63.2%
			In a stable relationship	81	12.4%
**Children**			Yes	358	54.8%
			No	295	45.2%
**Working categories**			Doctors/Physicians	286	43.8%
			Residents in training	99	15.2%
			Nurses	137	21.0%
			Others *	131	20.1%
**Positivity to COVID-19 swab**			No	564	86.4%
			Yes	89	13.6%
**COVID-19 related symptoms**			No	528	80.9%
			Yes	125	19.1%
**Health problems not related to COVID-19**			No	556	85.1%
			Yes	97	14.9%
**Changing of mansion due to the COVID-19 pandemic**			No	89	13.6%
			Yes	564	86.4%
**Job modification due to the COVID-19 pandemic**			No	322	49.3%
			Yes	331	50.7%
**Someone of dear positive to COVID-19 swab**			No	454	69.5%
			Yes	199	30.5%
**Modification of family habits for fear of infecting loved one dear**			No	43	6.6%
			Yes	525	80.4%
			No answer	85	13.0%
**Cut-off Scoring Questionnaires**					
**MHBI-HSS MP**			**High**	**Moderate**	**Low**
**EE**			>30	18–29	<17
**D**			>12	06–11	<5
**PA**			<34	35–39	>40
	**Severe**		**Moderate**	**Mild**	**Subclinical**
**IES TOT**	>44		26–43	9–25	0–8
			**Several Problems**	**Some Problemes**	**No problems**
**GHQ-12 TOT**			20–36	15–19	0–14
			**High**	**Moderate**	**Low**
**BAI TOT**			>36	22–35	0–21
	**Severe**		**Moderate**	**Mild**	**Minimal**
**BDI-II TOT**	29–63		20–28	14–19	0–13

*n* = number of participants. % = percentage of individuals. * = psychologists, socio-health, psychological, radiological and laboratory technicians, educators. Medical Personnel (MBI-HSS MP), Emotional Exhaustion (EE), Depersonalization (DP), Personal Accomplishment (PA), Beck Depression Inventory (BDI-II), Beck Anxiety Inventory (BAI), Impact of Event Scale (IES) and General Health Questionnaire 12 Items (GHQ-12).

**Table 2 ijerph-18-13083-t002:** The model implied (fitted) correlation matrix among Medical Personnel (MBI-HSS MP), Beck Depression Inventory (BDI), Beck Anxiety Inventory (BAI), Impact of Event Scale (IES) and General Health Questionnaire (GHQ), Age categories, Gender, COVID-19 related symptoms, Changing of mansion due to the COVID-19 pandemic.

Variables	Maslach Total	BAI Total	BDI-II Total	IES Total	GHQ-12 Total	Age Categories	Gender
Maslach Total	1.000						
BAI Total	0.760	1.000					
BDI Total	0.786	0.912	1.000				
IES Total	0.749	0.872	0.848	1.000			
GHQ Total	0.660	0.734	0.775	0.701	1.000		
Age categories	−0.324	−0.297	−0.233	−0.045	−0.156	1.000	
Gender	0.517	0.656	0.576	0.855	0.458	−0.023	1.000
COVID−19 related symptoms	0.168	0.520	0.495	0.319	0.397	0.025	0.020
Changing of mansion due to the COVID-19 pandemic	0.716	0.466	0.582	0.502	0.538	−0.091	0.147

**Table 3 ijerph-18-13083-t003:** Model implied (fitted) among Emotional Exhaustion (EE), Depersonalization (DP), Personal Accomplishment (PA), Beck Depression Inventory (BDI-II), Beck Anxiety Inventory (BAI), Impact of Event Scale (IES) and General Health Questionnaire 12 Items (GHQ-12), Age categories, Gender, COVID-19 related symptoms, Changing of mansion due to the COVID-19 pandemic.

Variables	EE	DP	PA	BAI Total	BDI-II Total	IES Total	GHQ-12 Total	Age Categories	Gender	COVID-19 Related Symptoms	Changing of Mansion due to the COVID-19 Pandemic
EE	1.000										
DP	0.360	1.000									
PA	0.228	0.079	1.000								
BAI Total	0.803	0.286	0.302	1.000							
BDI-II Total	0.806	0.316	0.289	0.912	1.000						
IES Total	0.843	0.108	0.219	0.873	0.848	1.000					
GHQ-12 Total	0.667	0.249	0.226	0.733	0.774	0.702	1.000				
Age categories	−0.187	−0.445	−0.217	−0.297	−0.233	−0.045	−0.156	1.000			
Gender	0.698	−0.106	0.164	0.656	0.576	0.855	0.458	−0.023	1.000		
COVID-19 related symptoms	0.210	0.136	0.220	0.520	0.495	0.319	0.397	0.025	0.020	1.000	
Changing of mansion due to the COVID-19 pandemic	0.556	0.372	0.057	0.466	0.582	0.502	0.538	−0.091	0.147	−0.010	1.000

**Table 4 ijerph-18-13083-t004:** Regression, covariance, intercept, and variance of Maslach Burnout Inventory—Human Services Survey for Medical Personnel (MBI-HSS MP), Beck Depression Inventory (BDI), Beck Anxiety Inventory (BAI), Impact of Event Scale (IES) and General Health Questionnaire (GHQ).

Variables			Estimate	Standard Error	Z-Value	P (>|z|)	Std. lv	Std. All
**Regression**	Maslach Total	Age categories	−0.835	0.059	−14.187	<0.001	−0.835	−0.261
		Gender	1.894	0.085	22.357	<0.001	1.894	0.414
		COVID-19 related symptoms	1.038	0.110	9.427	<0.001	1.038	0.173
		Changing of mansion due to the COVID-19 pandemic	3.936	0.116	33.995	<0.001	3.936	0.633
	BAI Total	Age categories	−1.259	0.059	−21.380	<0.001	−1.259	−0.265
		Gender	1.894	0.085	22.357	<0.001	1.894	0.414
		COVID-19 related symptoms	1.038	0.110	9.427	<0.001	1.038	0.173
		Changing of mansion due to the COVID-19 pandemic	3.936	0.116	33.995	<0.001	3.936	0.633
	BDI-II Total	Age categories	−0.760	0.059	−12.901	<0.001	−0.760	−0.190
		Gender	2.792	0.085	32.957	<0.001	2.792	0.488
		COVID-19 related symptoms	3.716	0.110	33.746	<0.001	3.716	0.495
		Changing of mansion due to the COVID-19 pandemic	3.878	0.116	33.500	<0.001	3.878	0.498
	IES Total	Age categories	−0.001	0.059	−0.018	<0.001	−0.001	−0.000
		Gender	9.650	0.085	113.896	<0.001	9.650	0.791
		COVID-19 related symptoms	4.908	0.110	44.576	<0.001	4.908	0.306
		Changing of mansion due to the COVID-19 pandemic	6.453	0.116	55.741	<0.001	6.453	0.389
	GHQ-12 total	Age categories	−0.273	0.059	−4.636	<0.001	−0.273	−0.115
		Gender	1.283	0.085	15.145	<0.001	1.283	0.377
		COVID-19 related symptoms	1.775	0.110	16.122	<0.001	1.775	0.397
		Changing of mansion due to the COVID-19 pandemic	2.208	0.116	19.074	<0.001	2.208	0.476
**Covariance**	Maslach Total	BAI Total	0.175	0.037	4.690	<0.001	0.175	0.175
		BDI-II Total	0.185	0.037	5.001	<0.001	0.185	0.185
		IES Total	0.128	0.038	3.358	0.001	0.128	0.128
		GHQ-12 total	0.067	0.039	1.735	0.083	0.067	0.067
	BAI Total	BDI-II Total	0.389	0.031	12.552	<0.001	0.389	0.389
		IES Total	0.232	0.036	6.457	<0.001	0.232	0.232
		GHQ-12 total	0.121	0.038	3.191	0.001	0.121	0.121
	BDI-II Total	IES Total	0.218	0.036	6.013	<0.001	0.218	0.218
		GHQ-12 total	0.245	0.036	6.867	<0.001	0.245	0.245
	IES Total	GHQ-12 total	0.076	0.039	1.957	0.050	0.076	0.076
**Intercept**	Maslach total		57.394	0.217	264.068	<0.001	57.394	26.887
	BAI Total		2.789	0.217	12.834	<0.001	2.789	0.878
	BDI-II Total		2.903	0.217	13.355	<0.001	2.903	1.086
	IES total		−1.512	0.217	−6.957	<0.001	−1.512	−0.265
	GHQ-12 Total		15.725	0.217	72.350	<0.001	15.725	9.889
**Variance**	Maslach total		1.000				1.000	0.219
	BAI Total		1.000				1.000	0.099
	BDI-II Total		1.000				1.000	0.140
	IES total		1.000				1.000	0.031
	GHQ-12 Total		1.000				1.000	0.395

Legend. Standardized latent variable coefficient (std.lv), Standardized coefficient (std. all).

**Table 5 ijerph-18-13083-t005:** Regression and covariance, intercept of Emotional Exhaustion (EE), Depersonalization (DP), Personal Accomplishment (PA), Beck Depression Inventory (BDI), Beck Anxiety Inventory (BAI), Impact of Event Scale (IES) and General Health Questionnaire 12 Items (GHQ).

Variables			Estimate	Standard Error	Z-Value	P (>|z|)
**Regression**	Emotional Exhaustion (EE)	Age categories	−0.0399	0.059	−6.774	<0.001
		Gender	2.702	0.085	31.888	<0.001
		COVID-19 related symptoms	1.167	0.110	10.596	<0.001
		Changing of mansion due to the COVID-19 pandemic	2.673	0.116	23.092	<0.001
	Depersonalization (DP)	Age categories	−0.785	0.059	−13.337	<0.001
		Gender	−0.459	0.085	−5.413	<0.001
		COVID-19 related symptoms	0.540	0.110	4.904	<0.001
		Changing of mansion due to the COVID-19 pandemic	1.314	0.116	11.353	<0.001
	Personal Accomplishment (PA)	Age categories	−0.349	0.059	−5.925	<0.001
		Gender	0.349	0.085	4.116	<0.001
		COVID-19 related symptoms	0.669	0.110	6.072	<0.001
		Changing of mansion due to the COVID-19 pandemic	0.052	0.116	0.448	0.654
	BAI Total	Age categories	−1.259	0.059	−21.380	<0.001
		Gender	3.987	0.085	47.058	<0.001
		COVID-19 related symptoms	4.630	0.110	42.051	<0.001
		Changing of mansion due to the COVID-19 pandemic	3.336	0.116	28.815	<0.001
	BDI-II Total	Age categories	−0.769	0.059	−12.901	<0.001
		Gender	2.792	0.085	32.958	<0.001
		COVID-19 related symptoms	3.716	0.110	33.747	<0.001
		Changing of mansion due to the COVID-19 pandemic	3.878	0.116	33.500	<0.001
	IES Total	Age categories	−0.001	0.059	−0.016	0.987
		Gender	9.650	0.085	113.896	<0.001
		COVID-19 related symptoms	4.908	0.110	44.576	<0.001
		Changing of mansion due to the COVID-19 pandemic	6.453	0.116	55.740	<0.001
	GHQ-12 total	Age categories	−0.273	0.059	−4.638	<0.001
		Gender	1.283	0.085	15.145	<0.001
		COVID-19 related symptoms	1.775	0.110	16.122	<0.001
		Changing of mansion due to the COVID-19 pandemic	2.208	0.116	19.072	<0.001
**Covariance**	Emotional Exhaustion (EE)	Depersonalization (DP)	0.294	0.034	8.560	<0.001
		Personal Accomplishment (PA)	0.055	0.039	1.423	0.155
		BAI Total	0.234	0.036	6.519	<0.001
		BDI-II Total	0.271	0.035	7.758	<0.001
		IES Total	0.122	0.038	3.197	0.001
		GHQ-12 total	0.111	0.038	2.900	0.004
	Depersonalization (DP)	Personal Accomplishment (PA)	−0.051	0.039	−1.301	0.193
		BAI Total	0.100	0.038	2.616	0.009
		BDI-II Total	0.102	0.038	2.653	0.008
		IES Total	0.041	0.039	1.042	0.297
		GHQ-12 total	0.012	0.039	0.317	0.751
	Personal Accomplishment (PA)	BAI Total	0.047	0.039	1.198	0.231
		BDI-II Total	0.088	0.039	2.266	0.023
		IES Total	−0.004	0.039	−0.106	0.916
		GHQ-12 total	0.042	0.039	1.080	0.280
	BAI Total	BDI-II Total	0.390	0.031	12.607	<0.001
		IES Total	0.237	0.036	6.617	<0.001
		GHQ-12 total	0.119	0.038	3.125	0.002
	BDI-II Total	IES Total	0.223	0.036	6.178	<0.001
		GHQ-12 total	0.239	0.036	6.671	<0.001
	IES Total	GHQ-12 total	0.077	0.039	2.003	0.045

**Table 6 ijerph-18-13083-t006:** Intercept and variance of Emotional Exhaustion (EE), Depersonalization (DP), Personal Accomplishment (PA), Beck Depression Inventory (BDI), Beck Anxiety Inventory (BAI), Impact of Event Scale (IES) and General Health Questionnaire (GHQ).

Variables		Estimate	Standard Error	Z-Value	P (>|z|)	Std. lv	Std. all
**Intercept**	Emotional Exhaustion (EE)	13.465	0.217	61.953	<0.001	13.465	6.662
	Depersonalization (DP)	12.200	0.217	56.131	<0.001	12.200	9.761
	Personal Accomplishment (PA)	68.272	0.217	314.117	<0.001	68.272	63.953
	BAI Total	2.789	0.217	12.834	<0.001	2.790	0.878
	BDI Total	2.903	0.217	13.355	<0.001	2.903	1.086
	IES total	−1.512	0.217	−6.957	<0.001	−1.512	−0.265
	GHQ Total	15.725	0.217	72.352	<0.001	15.725	9.890
**Variance**	Emotional Exhaustion (EE)	1.000				1.000	0.245
	Depersonalization (DP)	1.000				1.000	0.640
	Personal Accomplishment (PA)	1.000				1.000	0.877
	BAI Total	1.000				1.000	0.099
	BDI Total	1.000				1.000	0.140
	IES total	1.000				1.000	0.031
	GHQ Total	1.000				1.000	0.396

Legend. Standardized latent variable coefficient (std.lv), Standardized coefficient (std. all).

## Data Availability

The raw data supporting the conclusions of this article will be made available by the authors, without undue reservation.

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
