# Peer review of "The Mediating Role of Gender, Age, COVID-19 Symptoms and Changing of Mansion on the Mental Health of Healthcare Workers Operating in Italy during the First Wave of the COVID-19 Pandemic"

_ijerph, 2021, doi:10.3390/ijerph182413083_

Round 1

Reviewer 1 Report

In this study, the authors assess if there exists a possible relationship between work exhaustion and perceived stress, anxiety and depression among health professionals. Furthermore, they evaluate if the socio-demographic and anamnestic characteristics could act as a mediator in the relationship between work burnout and perceived stress, anxiety and depression. Unfortunately, their SEM model is ill-defined and denote ignorance about some basic notions of statistics. As an example they include categorical variables in the SEM and calculate Pearson correlations of categorical variables.

Therefore, I do not recommend acceptance of this manuscript for publication.

Author Response

We agree with the reviewer's concerns; in this regard, some points that might have raised methodological doubts for the reader have been better specified in the method section.
All endogenous dependent variables are quantitative; the Pearson correlation matrix with associated densities and histograms have been reported for these variables. The dichotomous variables used in the SEM model (gender, change in habits during the pandemic, etc… ) are exogenous (independent).
Those variables have been recorded as a dummy (0/1) variable like in a classic regression model as suggested in the literature (Rosseel 2014). The age in classes variable is ordinal and exogenous the encoding scheme reflects the order (say, 1,2,3,. . . ) and has been treated like any other (numeric)
covariate (Rosseel 2014).
The Pearson’s correlation has been reported for estimating the correlation across continuous
variables; the biserial (polyserial) correlations for the continuous and binary (ordinal) variables, and Tetrachoric (or Polycoric) correlation for binary (ordinal) variables.
The specification has been included in the method section.

Reviewer 2 Report

In this paper the authors describes the results of an investigation on Heath Care Workers (HCWs) during the first wave of COVID 19 in Italy to assess the influence on mental health in particular relating to relationship between job burnout, gender, impact of pandemic and changing of mansion with depression, psychological distress, health perception, anxiety and distress. The obtained results revealed correlations among these parameters, underlining the importance to adopt specific educational strategies to support HCWs in these particular situations.

The topic and the results are very interesting relating to the impact of COVID 19 pandemic situation and they could be used as basis for increase the interventions.

However, to be published, the manuscript must be improved:

The entire text must be revised for English language and for terminology: i.e. LINE 145: “…2019-nCOVID…” is not correct, the disease calls COVID 19, while the etiological agent is SARS CoV 2.

Introduction must be improved because several parts are unclear: I suggest to reduce it, to describe very well the state of the art and the objective of the research.

Specific suggestion

LINE 57: “….364,749 recovered cases in Italy in the last 30 days (EpiCentro, 2021)…”: what is the times referred to 30 days?

Methods, is well organized.

Results and Discussion

Specific suggestion

LINE 204: Only 37% of HCWs filled the survey: the authors could explain in discussion what is the reason of this loss of data?

Author Response

In this paper the authors describe the results of an investigation on Heath Care Workers (HCWs) during the first wave of COVID 19 in Italy to assess the influence on mental health in particular relating to relationship between job burnout, gender, impact of pandemic and changing of mansion with depression, psychological distress, health perception, anxiety and distress. The obtained results revealed correlations among these parameters, underlining the importance to adopt specific educational strategies to support HCWs in these particular situations.

The topic and the results are very interesting relating to the impact of COVID 19 pandemic situation and they could be used as basis for increase the interventions.

However, to be published, the manuscript must be improved:

The entire text must be revised for English language and for terminology: i.e. LINE 145: “…2019-nCOVID…” is not correct, the disease calls COVID 19, while the etiological agent is SARS CoV 2.

Thank you for your comment. Regarding professional editing, we will certainly be willing to perform it as soon as we have a definitive version of the paper, considered suitable by all reviewers.Thank you for the suggestion, concerning the terminology of 2019-nCOVID, we corrected it inserting SARS CoV 2.

Introduction must be improved because several parts are unclear: I suggest to reduce it, to describe very well the state of the art and the objective of the research.

We tried to reorganize the introduction following your precious indications.

Specific suggestion

LINE 57: “….364,749 recovered cases in Italy in the last 30 days (EpiCentro, 2021)…”: what is the times referred to 30 days?

We corrected the sentence presents in line 57, specifying data determined on 18 November 2021.

Methods, is well organized.

Results and Discussion

Specific suggestion

LINE 204: Only 37% of HCWs filled the survey: the authors could explain in discussion what is the reason of this loss of data?

According to your suggestion, we inserted an explanation about the loss of data in line 204.

Reviewer 3 Report

The article presenting the research should be evaluated highly. It provides a new insight into the mental health of healthcare professionals and shows related risk factors. An additional advantage of the research is the fact that it draws attention to mediation factors and provides an in-depth analysis. The selection of samples, research methods and the discussion of the results are correct. Only the lack of the characteristics of the independent variables deserves attention. Maybe it's worth changing. Instead of the description of the variables in lines 204-223, the values ​​of which are in Table 1, it is worth presenting the dependent variables: (Maslach Burnout Inventory - Human Services Survey for Medical Personnel-MBI-HSS MP), overall health perception (General Health Questionnaire-12 Items-GHQ-12), distress perceived because of stressful life events (Impact of Event Scale-IES), depression (Beck Depression Inventory-BDI-II) and anxiety (Beck Anxiety Inventory-BAI). Showing the values ​​of the variables will allow the reader to understand the issue better.

Author Response

We really appreciate your suggestions. We inserted in the methods a description of dependent variables, according to your suggestions, maintaining also the description of independent variables. Moreover, specifications referring to the suggested variables have been added in Table 1 and the results.

Round 2

Reviewer 1 Report

In this study, the authors assess if there exists a possible relationship between work exhaustion and perceived stress, anxiety and depression among health professionals. Furthermore, they evaluated if the socio-demographic and anamnestic characteristics could act as a mediator in the relationship between work burnout and perceived stress, anxiety and depression.

It is not clear what the authors mean about changes in job tasks and duties, they should better clarify this concept.

The sentence in lines 57-62 should be rephrased.

The acronyms EE, D and PA in background section should be firstly mentioned in the extended version and then in the remaining manuscript the authors could use directly the acronyms.

The aim is repeated twice.

It is not clear the terms “second median and third quartile” maybe the authors should change with median and interquartile range (Q1-Q3).

It was very difficult to me to understand the results because they do not flow at all. So, I will need to read them again after the authors follow these suggestions:

There is confusion between Figure 2 and Table 2 and between Figure 3 and Table 3. In fact, they seem to report the same correlations, but coefficients are absolutely different. You must clarify what is the difference between them in the titles.

The authors should move the possible explanation of low response rates to the discussion section.

In line 239 maybe the coefficient of perceived stress on depression should be 0.58.  

In line 383 the coefficient of EE and depression is missing and the coefficient of EE and health perception should be 0.46.

In lines 417-418 health perception and its coefficient is repeated twice.

In Fig 3. Some labels are missing please correct it.

There are too many tables and figures. Maybe the tables could be unified trying to delete the redundant variables.

The quality of the figures should be improved.

Please move the comment of the model from the discussion to the results section.

The discussion should include the following: Summary of results (without reporting the coefficients) and Comparison with other studies; Strengths & limitations; Conclusion etc.

Author Response

In this study, the authors assess if there exists a possible relationship between work exhaustion and perceived stress, anxiety and depression among health professionals. Furthermore, they evaluated if the socio-demographic and anamnestic characteristics could act as a mediator in the relationship between work burnout and perceived stress, anxiety and depression.

Thank you for the suggestion, we tried to follow the suggestions of reviewer 1 that we thank him for his precious support and the review work.

It is not clear what the authors mean about changes in job tasks and duties, they should better clarify this concept.

We tried to clarify what we mean about changes in job tasks and duties, following your suggestions.

The sentence in lines 57-62 should be rephrased.

We rephrased lines 57-62

The acronyms EE, D and PA in background section should be firstly mentioned in the extended version and then in the remaining manuscript the authors could use directly the acronyms.

Thank you for the tips. We firstly mentioned acronyms EE, D, and PA in the extended version in the background section and then we used directly the acronyms, according to what you said.

The aim is repeated twice.

We rephrased the aim, removing repetitions.

It is not clear the terms “second median and third quartile” maybe the authors should change with median and interquartile range (Q1-Q3).

Thank you for the suggestions. In the method section, we changed “second median and third quartile” with median and interquartile range (Q1-Q3).

It was very difficult to me to understand the results because they do not flow at all. So, I will need to read them again after the authors follow these suggestions:

There is confusion between Figure 2 and Table 2 and between Figure 3 and Table 3. In fact, they seem to report the same correlations, but coefficients are absolutely different. You must clarify what is the difference between them in the titles.

We clarified what is the difference between figure 2 and Table 2 and between Figure 3 and Table 3 in the titles.

The authors should move the possible explanation of low response rates to the discussion section.

We moved the possible explanation of low response rates to the discussion section.

In line 239 maybe the coefficient of perceived stress on depression should be 0.58.  

In line 383 the coefficient of EE and depression is missing and the coefficient of EE and health perception should be 0.46.

In lines 417-418 health perception and its coefficient is repeated twice.

We made corrections as you indicated in line 239, line 383, lines 417-41.

In Fig 3. Some labels are missing please correct it.

We corrected them.

There are too many tables and figures. Maybe the tables could be unified trying to delete the redundant variables.

We agree with the reviewer, the labels have been rearranged; the figures report the observed Pearson correlations, and the tables report the model implied (fitted) correlations

The quality of the figures should be improved.

We improved the quality of figures.

Please move the comment of the model from the discussion to the results section.

The discussion should include the following: Summary of results (without reporting the coefficients) and Comparison with other studies; Strengths & limitations; Conclusion etc.

We rephrased the discussion removing the coefficient by the results, maintaining comparison with other studies; we added, as you suggested, Strengths & limitations; Conclusion. We opted to not move the comment of the model from the discussion, which we consider to be more suitable for the discussion, rather than the result section.
